# Chitosan Functionalization: Covalent and Non-Covalent Interactions and Their Characterization

**DOI:** 10.3390/polym13234118

**Published:** 2021-11-26

**Authors:** Laura Nicolle, Céline M. A. Journot, Sandrine Gerber-Lemaire

**Affiliations:** Group for Functionalized Biomaterials, Institute of Chemical Sciences and Engineering Ecole Polytechnique Fédérale de Lausanne, EPFL SB ISIC SCI-SB-SG, Station 6, CH-1015 Lausanne, Switzerland; laura.nicolle@epfl.ch (L.N.); celine.journot@epfl.ch (C.M.A.J.)

**Keywords:** chitosan, functionalization, chemical modifications, covalent conjugation, electrostatic interactions, characterization

## Abstract

Chitosan (CS) is a natural biopolymer that has gained great interest in many research fields due to its promising biocompatibility, biodegradability, and favorable mechanical properties. The versatility of this low-cost polymer allows for a variety of chemical modifications via covalent conjugation and non-covalent interactions, which are designed to further improve the properties of interest. This review aims at presenting the broad range of functionalization strategies reported over the last five years to reflect the state-of-the art of CS derivatization. We start by describing covalent modifications performed on the CS backbone, followed by non-covalent CS modifications involving small molecules, proteins, and metal adjuvants. An overview of CS-based systems involving both covalent and electrostatic modification patterns is then presented. Finally, a special focus will be given on the characterization techniques commonly used to qualify the composition and physical properties of CS derivatives.

## 1. Introduction

Chitosan (CS) is a natural biopolymer composed of repeating β-(1,4)-2-amino-D-glucose and β-(1,4)-2-acetamido-D-glucose units that are linked by 1,4-β-glycosidic bonds (Figure 1). The industrial production of CS relies on the partial deacetylation of chitin, a polymer widely present in crustacea’s shell and fungi [1,2].

Due to its biocompatibility, biodegradability, versatility, and low price, CS has gained a lot of attention over the past decades in fields ranging from wound healing and drug delivery [3,4,5] to waste water treatment [6,7], textile industry [8], and food packaging [9]. The versatility of CS relies on its amino and hydroxyl groups (Figure 1) enabling various types of functionalization that will be described in detail in this review. Figure 2 summarizes the different strategies to modify CS using either covalent conjugations or non-covalent interactions.

CS is characterized by its molecular weight (MW), degree of deacetylation (DD), and polydispersity index (PDI). CS exists over a wide range of MW and DD, which are generally selected according to the targeted application. Very low MW CS chains (<3900 Da usually, <20 units, named CS oligosaccharides (COS)) are known for their enhanced solubility in aqueous and polar media compared to higher MW CS [10]. The MW is considered one of the main factors impacting CS solubility. Other parameter such as DD, purity of the batch, and pattern of acetylation can also influence the solubility [11,12]. Poor characterization of CS is strongly impairing its use in interested research fields, highlighting the need to standardize characterization procedures [12]. Many suppliers of commercial CS batches do not refer to MW but only indicate viscosity value or MW range (“high”, “medium”, or “low” MW CS).

The development of CS derivatives is greatly impaired by their poor solubility for which depolymerization is one of the main strategies to improve it. Degradation methods include enzymatic treatment [13,14] and chemical depolymerization under oxidative [15], basic [16] or acidic [17] conditions. Enhancement of CS solubility is otherwise achieved by functionalization, such as with trimethyl CS (TMC) or O-carboxymethyl CS (CMCS). These techniques will be further discussed in the next sections.

This review aims at presenting the recent methodologies that have been implemented for CS functionalization, focusing on the last five years of literature in order to reflect at best the state-of-the-art on CS derivatization. Numerous reviews have already described the production and properties of CS [1,18,19], its fields of application [20,21], as well as specific modification patterns depending on the targeted applications [22,23]. We herein concentrate on the functionalization strategies and the characterization techniques for CS derivatives.

This review will first describe covalent functionalizations on the CS backbone, its non-covalent interactions with small molecules, proteins, or metals, and finally, the combination of both types of modifications to reach more potent carriers or materials. Special focus will also be given to the characterization techniques that are commonly used to qualify the composition and physical properties of CS derivatives.

In order to clarify the terminology used for CS functionalization, linkers will refer to molecular entities enabling the covalent conjugation of CS to other components (small molecules or polymers), whereas cross-linkers will refer to binding entities between CS chains.

## 2. Strategies for Covalent Functionalization

The common methodologies for CS covalent modification rely on chemical derivatization of the primary amine and alcohol functionalities, or partial oxidative cleavage of the backbone. CS thiolation was also reported for several targeted applications and was recently reviewed by Federer et al. and Summonte et al. [24,25]. While CS is most often used as starting material, more advanced intermediates such as CMCS, TMC, or O-glycol CS (GCS) are also commercially available to facilitate downstream modifications (Figure 3). Of note, reacetylation of CS and, more specifically, GCS has also been reported in order to obtain O-glycol chitin [26].

### 2.1. Amine Functionalization

#### 2.1.1. Acylation Leading to Amide Bonds

CS is most frequently functionalized with other polymers or small molecules via formation of amide bonds in the presence of coupling agents or anhydride substrates. The resulting conjugates demonstrate high stability towards most biological events.

##### Amide Bond Formation in the Presence of Coupling Agents

The most classical way to functionalize amines of CS is to activate the carboxylic acid derivative of interest with a coupling agent and then acylate the amines of CS with the newly formed reactive intermediate (Figure 1). Due to its favorable solubility in water, N-(3-Dimethylaminopropyl)-N’-ethylcarbodiimide chloride (EDC.HCl) is the coupling agent of choice [27]. However, its stability in aqueous solutions being limited, its use in polar solvents such as DMF or N-methyl pyrrolidone (NMP) is often preferred when possible.

Most reports present the use of EDC.HCl in combination with N-hydroxysuccinimide (NHS) or hydroxybenzotriazole (HOBt) to improve the stability of the activated intermediates prior to conjugation with CS amines [28,29,30,31,32]. Some substrates of interest are also supplied as NHS ester derivatives, thus facilitating their coupling with CS. For instance, Chong et al. used methoxyPEG succinimidyl succinate (mPEG-ss) and palmitic NHS ester (PNS) for the preparation of hydrophobic GCS derivatives [33]. The resulting amphiphilic micelles formed in aqueous solution were applied to the encapsulation of the hydrophobic drug itraconazole. Sustained and high cumulative drug release was observed, suggesting those micelles as promising nanocarriers for the delivery of hydrophobic bioactive ingredients.

Numerous examples of carboxylic activation were disclosed in the literature, and some relevant ones are listed at the end of this section in Table 1. In addition, it is worth mentioning that the conjugation of CS amines to activated carboxylic acid derivatives can be combined with post-functionalization cross-linking processes. Elsayed et al. reported the preparation of CS-based membranes through the coupling of CS with activated *trans*-3-(4-pyridyl) acrylic acid, followed by covalent cross-linking of the pyridyl moieties via UV light promoted [2+2] cycloaddition [34]. Similarly, the conjugation of CS amines to activated 5-norbornene-2-carboxylic acid allowed for subsequent inverse electron-demand Diels Alder cycloaddition in the presence of a bis-tetrazine-containing cross-linker to afford hydrogels with increased pore size [35]. This system demonstrated enhanced capacity for the loading and further release of 5-aminosalicylic acid, in comparison with the reference glutaraldehyde (Glu) cross-linked CS hydrogels.

Although quite rare, other activation techniques using 4-nitrophenyl chloroformate (4-NPC) [36] or succinimidyl 3-(2-pyridyldithio)propionate (SPDP) [37] were also reported.

##### Amide Bond Formation in the Presence of Anhydride Substrates

Along with the activation mediated by coupling agents, anhydrides have long been reported in literature for conjugation to CS amines under mild conditions [38,39,40].

With the exception of methacrylic anhydride, all reported anhydrides for CS functionalization are cyclic, which allow further functionalization with amino-based systems after ring-opening (Figure 4). This strategy was frequently reported for the conjugation to small molecules and polymers such as branched and linear polyethylenimines (B/LPEI), leading to gene delivery vehicles [39,41].

Succinic anhydride is the most often reported cyclic anhydride for CS derivatization, affording N-succinyl CS (NSCS), which presents improved water solubility over a wide range of pH (acidic to basic ones), high hydrophilicity, and sustained biocompatibility compared to native CS [42]. While the improved water solubility of NSCS enables further functionalization with Gum Arabic multiple aldehyde [43], its anionic property was exploited in the preparation of non-covalent systems, such as layer-by-layer liposome coating in the work of Seong et al. [44]. Additional examples are mentioned in Table 1.

The reverse synthetic pathway was also reported in the literature: the substrate of interest was first derivatized with an anhydride, followed by the grafting of the resulting system onto amino groups of CS via EDC.HCl/NHS or similar reagents [45].

##### Selected Examples of CS Functionalization through Amide Bond Formation

The use of either coupling agents or anhydride substrates is equally reported in the literature for CS functionalization. Table 1 presents selected and relevant references using these methodologies with the field of application of the resulting final systems.

**Table 1 polymers-13-04118-t001:** Amide bond formation on CS systems and applications of the resulting conjugates.

Method	Grafted Substrates	Application	Ref.
**EDC.HCl (+/− NHS)**	CSK peptide	Polymeric nanocarrier to promote drug absorption and oral bioavailability in anticancer therapy	[46]
	Folic acid	Polymeric NPs for tumor targeted drug delivery	[28]
	SWCN	NPs for mediated-chloroplasts transgene delivery	[47]
	DEACMS	CS-coumarin-derived micelles for pesticide release	[45]
	GBA; LA	Targeted delivery of DOX to CXCR 4 tumor cells	[48]
	HCA	Surgical adhesive	[49]
	OA; GA	MRI guided theranostic cancer therapy	[30]
	4-pentynoic acid	Food packaging material with improved shelf-life	[50]
	PBA-COOH	Curcumin encapsulation and ROS-triggered drug release	[51]
	PCA	Hydrogels for antioxidant material in drug release and tissue engineering	[31]
**Anhydride**	Methacrylic anhydride	CS-silk fibroin hydrogels for wound dressing	[52]
	Maleic anhydride	CS-BPEI-Arg NPs for gene therapy	[39]
	Succinic anhydride	Gene delivery vehicle	[41]
**EDC.HCl and** **anhydride**	Succinic, phthalic, glutaric, and phenylsuccinic anhydrides; TPP	Polymeric NPs for drug delivery	[40]
	Succinic anhydride; Brij-S20	Intestinal absorption enhancement of 123-Rhodamine (Pgp substrate)	[32]
	Succinyl prednisolone; succinic anhydride	Nanogels for treatment of ulcerative colitis	[53]

Abbreviations: CSK peptide: CSKSSDYQC; NP: Nanoparticle; SWCN: Single-walled carbon nanotube; DEACMS: 7-diethylaminocoumarin-4-yl methyl succinate; GBA: Guanidino benzoic acid; LA: lactobionic acid; DOX: Doxorubicin; CXCR 4: C-X-C chemokine receptor type 4; HCA: Hydrocaffeic acid; OA: octadecanoic acid; GA: Gadopentetic acid; MRI: Magnetic resonance imaging; PBA-COOH: 3-carboxyphenylboronic acid; ROS: Reactive oxygen species; PCA: protocatechuic acid; BPEI: branched polyethylenimine; Arg: Arginine; TPP: tripolyphosphate; Brij-S20 (trade name): polyoxyethylene (20) stearyl ether; Pgp: P-glycoprotein.

#### 2.1.2. Phthaloylation

Phthaloylation is a method of choice to protect CS amines while enhancing the solubility of the resulting intermediate. It is usually performed with phthalic anhydride in DMF at 120–130 °C (Figure 2).

Phthaloylation leads to increased solubility of the protected polymer in organic media [54]. The selectivity toward N-phthaloylation over protection of the hydroxyl groups can be enhanced by the addition of water to the reaction solvent [55]. However, it was observed that side O-phthaloylation, even in small proportions, led to increased solubility of Phth-CS in organic solvents, such as DMF, DMSO, and pyridine. The selective N-phthaloylation method was recently applied to the preparation of antifouling polyethersulfone membranes [56] and siRNA carriers based on an acid-transforming CS [57].

#### 2.1.3. Acylation Leading to Urea Bonds

Urea bond formation on CS amines is commonly reported in water or polar solvents in the presence of 1,1’-carbonyldiimidazole (CDI) as activating agent. Most pathways rely on a one-pot procedure to avoid the isolation of the sensitive acyl imidazole intermediate (Figure 3).

Despite the moisture sensitiveness of CDI, the use of slightly acidic aqueous solutions was reported for the derivatization of CS with cationic polymers such as low MW BPEI [58]. Under similar conditions, CS-BPEI copolymers were produced as non-viral gene carriers targeting chondrocytes and synoviocytes [59]. The conjugation of CS with BPEI-cystamine, in the presence of CDI, led to the formation of bioreducible copolymers for gene delivery [60]. While ^1^H NMR analysis of the final system gave evidence for the combination of CS and BPEI, one cannot disregard that the copolymers were assembled by non-covalent interactions rather than bis-urea bonds due to the fast degradation of CDI in aqueous medium (see Section 5 for more details on the characterization of covalent CS-based copolymers). Dry polar solvents should be preferred when using CDI, even though activation requires a longer time than with water [61].

Alternatively, isocyanate derivatives were also reported for the formation of urea bonds [62,63].

#### 2.1.4. Alkylation

Several reports describe the grafting of small aliphatic halides on CS amines via nucleophilic substitution (S_N_), leading to secondary amine linkages (Figure 5). For instance, bromination of geraniol (Ger) in the presence of PBr_3_, followed by conjugation to CS amines, was developed for the mass production of the potential food preservative COS-N-Ger [64] (Figure 5A). A similar pathway was followed with (–)-citronellol (Cit) [65] (Figure 5B). Dibromopropane was used for the conjugation of CS to melamine (Mel) [66] to afford CS-Pr-Mel as a bifunctional organocatalyst promoting Knoevenagel condensations (Figure 5C). Such a polymer-based catalyst offers several advantages over metallic species, including minimized pollution due to their biodegradability.

N-Methylation of CS is widely encountered for the preparation of quaternized CS, which are water-soluble systems at neutral and alkaline pH, thus enlarging the scope of CS applications [67,68]. TMC is one of the most well-known quaternary ammonium CS salts. While it is commercially available, several procedures reported the production of TMC in the presence of excess amounts of MeI and NaOH at 60 °C [37,46,69]. In particular, a two-step procedure was preferred to ensure complete CS N-trimethylation and a higher yield of the quaternized polymer [46,69].

#### 2.1.5. Epoxide and Aziridine Ring Openings

Epoxide ring opening of glycidol derivatives also applies to the functionalization of CS amines by S_N_ (Figure 4). The resulting derivatives were reported to form micelles, hydrogels, or membranes and are commonly used for drug/gene delivery.

Such reactions are generally performed in basic aqueous buffers to preserve the nucleophilicity of CS amines [70] and at controlled temperatures below 50 °C [62,71]. The resulting CS derivatives display enhanced solubility in aqueous and organic media, allowing further derivatization of the remaining free amines with activated polymers such as isocyanate terminated polycaprolcatone [62]. CS crosslinking was also reported in the presence of small epoxide derivatives such as glycidyl methacrylate, which can undergo both epoxide ring opening and 1,4-addition from CS amines [71] (Figure 4C). Similarly, the use of low MW polymeric cross-linkers such as diglyceryl PEG (PEGDGE, M_n_ 500 Da) resulted in CS-based chromatography affinity membranes [72] (Figure 4D). Glycidyltrimethylammonium chloride (2,3-epoxypropyltrimethylammonium chloride, EPTAC) is another glycidol derivative that has been reported for CS quaternization [73] (Figure 4E, see Section 4.1 for more details).

Moreover, aziridine ring opening was used for CS conjugation to PEI [74]. One of the proposed mechanisms is the actual polymerization of aziridine moieties on CS.

#### 2.1.6. 1,2-Nucleophilic Addition: Imine Formation

The condensation of CS amines to aldehydes was vastly investigated for CS functionalization (Figure 5). Due to the reversibility of this reaction, the resulting CS imines often find applications in self-healing polymeric derivatives [43,75].

While aliphatic imines are rather unstable, aromatic imines display higher stability, resulting from the possible charge delocalization on the aromatic ring. Condensation with benzaldehyde was thus used for the protection of CS amines and subsequent modification of the primary alcohols [76,77].

Besides protective purposes, imine bonds are often reported for the cross-linking of CS chains. Glu is the most common CS cross-linker, leading to the stabilization of CS nanoparticles (NPs) (Figure 6, right circle) [6,51]. More specifically, Glu can also be exploited as a simple linker for the conjugation of CS to another polymer containing amino groups, such as polyacrylamide block copolymer [78]. Squaric acid is another efficient cross linker leading to spontaneous formation of imine bonds with CS chains [79] (Figure 6, left circle). The resulting cross-linked CS gels presented significantly improved mechanical and swelling properties.

Polymers and small molecules bearing aldehydes functionalities also form imine bonds with CS amines. For example, selenium- and sulfur-containing derivatives of citronellal and citral [80], as well as aliphatic aldehydes ranging from butanal to octanal [81], were successfully grafted on CS. Resulting derivatives respectively displayed improved antioxidant activity and enhanced moisture stability. Aldehyde-containing polymers are chosen for their specific properties in relation to desired applications. For example, oxidized hyaluronic acid (HA) was grafted on CMCS to form a hydrogel for wound dressing purposes [82]. Both polymers displayed hemostatic properties, facilitating platelets aggregation and activating red blood cells. Alternatively, sodium alginate (alg) can be employed for wound healing applications due to its water-retaining property. Its combination with COS allowed enhancement of the mechanical properties compared to pure alg hydrogels [83]. Regarding drug delivery, Gum Arabic multiple aldehyde was used as a non-toxic macromolecular CS cross-linker [43].

Other examples of CS-based imine systems were disclosed from reaction with nitriles [75] or as intermediates of irreversible transformations including Mannich reaction [84] or reductive amination [85,86]. The resulting systems showed enhanced stability.

#### 2.1.7. 1,4-Nucleophilic Addition

Lastly, 1,4-nucleophilic additions of CS amines to α,β-unsaturated carbonyls were recently reported in the literature for covalent conjugation purposes. These aza-Michael reactions involve acrylate derivatives such as methyl acrylate [87], itaconic acid (IA) [88], or glycidyl methacrylate [71]. Interestingly, 1,4-addition of CS amines to IA was performed in aqueous solution at 90 °C, without the need for any promoter, leading to a di-carboxylic acid intermediate that spontaneously cyclized to the corresponding pyrrolidone derivative, with an overall grafting degree of 10% (Figure 6). The resulting polymer demonstrated excellent zwitterionic properties, making it applicable at any pH for water treatment.

### 2.2. Hydroxyl Functionalization

Both CS primary alcohol in C6 position and secondary alcohol in C3 position might undergo chemical derivatization (Figure 1). While no evidence was given yet for the regioselective modification of the primary alcohol, it is most probably the main site for hydroxyl functionalization due to its better accessibility. CS amines being more nucleophilic than the alcohols, N-protection or functionalization (detailed in Section 2.1) is required for selective hydroxyl functionalization.

#### 2.2.1. Acylation Leading to Carbamate Bonds

Similarly to urea bonds encountered for amines modifications, carbamate bonds ensure reliable, covalent, and durable linkages to CS alcohols. In presence of CDI as activating agent in dry solvent, both aliphatic amines and amino-terminated PEG derivatives were grafted to CS hydroxyl groups. Following this procedure, the conjugation to dodecyl amine led to CS-based self-assembled micelles for gene delivery [89]. Alternatively, CS-PEI copolymers were further conjugated to PEG derivatives through O-functionalization to evaluate the shielding effect of PEG chains on the targeted gene delivery vehicles [41].

#### 2.2.2. Acylation Leading to Ester Bonds

Due to their susceptibility to hydrolysis under physiological conditions, ester linkages were scarcely reported for the conjugation to CS alcohols. Zhu et al. described the O-succinylation of cross-linked CS for the preparation of functionalized membranes for metal affinity chromatography applications [72]. CS esters deriving from the reaction with maleic anhydride were further derivatized to produce CS-based thermo sensitive polymers designed for the coating of gold NPs [90]. Despite their reduced stability in vivo, in comparison to amide or carbamate functionalities, esters might be introduced in drug delivery carriers in cases where fast degradability is required.

#### 2.2.3. Transacetalisation

The innovative preparation of acid-transforming CS (ATC) was achieved by transacetalisation of CS hydroxyls with trifluoroacetamide-protected aminoethoxy ketal (TFA-AE-k) [57] (Figure 7). In comparison with native CS, ATC demonstrated increased bioavailability in aqueous media and improved molecular interactions with siRNA. Due to the acid-sensitivity of ketal linkages, siRNA condensed by ATC was efficiently released into the cytosol, thus supporting this strategy for gene silencing applications.

#### 2.2.4. Nucleophilic Substitution

As regards hydroxyls, nucleophilic substitutions are mostly performed on halogenated derivatives. It includes alkylation with small molecules [76], carboxymethylation [31] and tosylation, which allowed in this case further transformation into terminal azides [91]. Interestingly, azido-CS derivatives were further functionalized through a copper-catalyzed click reaction with TEMPO-PEO-alkyne [91].

Despite being commercially available, in-house synthesis of CMCS remains frequent [82]. The initial procedure developed by Liu et al. in 2000 involved the nucleophilic substitution of monochloroacetic acid in aqueous NaOH/*^i^*PrOH, which favored the selective O-carboxymethylation [92]. Other parameters such as low temperature [93] or careful pH control [94] were reported to influence the selectivity between O- and N-carboxymethylation. Apart from applications in cancer therapy research [93,95], CMCS was recently used in more diverse applications, such as hemostatic material [82] or antioxidant hydrogel for drug release or tissue engineering [31].

In a similar fashion to the amino groups (Section 2.1.5), the hydroxyl groups of CS can also mediate ring opening of glycidol derivatives under basic conditions. The functionalization of CS alcohols with allyl glycidyl ether was used for a subsequent thiol-ene crosslinking reaction in the presence of thiol-terminated 4-arms PEG derivatives [77].

### 2.3. Oxidative Cleavage

The disruption of CS sugar units through oxidative cleavage leads to the formation of reactive aldehydes for downstream functionalization (Figure 8). Most reported oxidation pathways made use of periodate reagents (KIO_4_ or NaIO_4_) in degassed acetate buffer at 4 °C [96,97]. These conditions also result in extensive depolymerization due to the release of ammonia, which can be limited by higher deacetylation degrees of the starting polymer [96,98].

CS oxidation was often applied to further conjugation with PEIs through reductive amination, leading to the production of condensing copolymers for drug and gene delivery applications [97]. More complex systems involved subsequent conjugation of the PEI amines to TAT and cyclic RGD peptides to promote both the cellular uptake and targeting ability of the resulting polymeric vectors [99].

Keshk et al. also explored the oxidation of CS in neutral medium, revealing reduced side-depolymerization and unchanged DD [100]. However, the oxidation degree of CS was much lower.

## 3. Non-Covalent Modifications

CS-based materials have also been combined with small molecules, proteins, and polymers via non-covalent modifications. Under these conditions, the stability of the resulting CS derivatives relies on three main types of non-covalent interactions: H-bonding, electrostatic interactions (ionic bonds), and chelation (Figure 7A–C). Entanglement of larger molecules into a CS-based matrix was also reported and can add to the stabilization of the resulting hydrogel, as illustrated in Figure 7D.

The assembly of CS derivatives through non-covalent modifications avoids the need for chemical reagents and thus simplifies purification procedures, which can be complex and time-consuming. While non-covalent conjugates are easily prepared, there is generally little control over their association and interaction patterns. In addition, their chemical characterization remains very challenging and does not provide precise information on the intermolecular organization and availability of functional moieties (see Section 5 for more details on characterization). This section focuses on solid CS-based materials (nanofibers, NPs, and dry films) or hydrogels. Interestingly, the mechanical resistance and stability of CS derivatives can be significantly improved by their non-covalent association to specific additives. This strategy was highlighted in bone tissue engineering and wound dressing applications [101,102] and is also suitable for the development of pH-responsive conjugates [103]. This section describes systems involving non-functionalized CS forming non-covalent interactions with other components including polymers, proteins, small molecules, and nanocomposites.

### 3.1. Non-Covalent CS Conjugates with Proteins

The development of therapeutic proteins has attracted much interest in recent years [104]. Their activity is compromised in case their 3D structure is disrupted or their integrity is affected by undesired chemical reactions. Exposure to heat, change of pH, change of salt concentration, and strain are all susceptible to result in protein misfolding. Moreover, endogenous proteases are responsible for most of the inactivation processes of protein drugs in the body [105]. Once conjugated to CS through electrostatic interactions, proteins are stabilized and protected from exposure to chemical and physical strains. In addition, CS exerts other beneficial properties by its ability to open the tight junctions between epithelial cells and their significant mucosal adhesive capacity [105].

Furthermore, the development of a drug for oral delivery must face the extreme acidic condition of the digestive system. Therefore it is good practice to provide a study of the complexes’ stability in various conditions to determine the range of application [106,107]. For example, Prudkin Silva et al. protected insulin (I) from the low pH of the gut by forming nanocomplexes with CS [108]. They studied I-CS assemblies at different CS concentrations and pH by dynamic light scattering (DLS), zeta-potential, and absorbance. They observed that the isoelectric point (Ip) of I was a turning point in the macromolecular behavior. They concluded that at pH > Ip and high CS concentration (10^−2^% (*w*/*w*)), a core–shell complex formed, whereas at lower CS concentration (10^−3^–10^−4^% (*w*/*w*)), a bridging–flocculation process occurred, forming larger complexes (Figure 9). Finally, they showed that the use of CS prevented aggregation of the protein and could shield it for controlled release.

For the development of oral vaccines, researchers have reported that antigen-NP delivery triggers a stronger immune response compared to a naked antigen [109]. With the same observation, Wu et al. studied oral and intramuscular delivery of CS-coated silica NP loaded with BSA (as antigen model) [110]. They showed that the protein maintained its structure after release from the NPs, and that oral delivery was able to trigger a good immune response. It provides a promising road towards the development of a novel oral vaccine with significant advantages such as cost-effectiveness, patient safety, and compliance.

Whey proteins, on the other hand, are extremely resistant to heat and acidic conditions and can further improve the stability of a product or system. Whey–CS systems have been shown to increase resistance to oxidation and to emulsion stability by preventing flocculation and coalescence [111]. Consequently, whey–CS systems are commonly found in development of food packaging and additives for human consumption, as well as, occasionally, in drug delivery systems [111,112,113,114].

### 3.2. Non-Covalent CS Conjugates with Other Biopolymers

Electrostatic interactions between CS and a negatively charged polymer form a polyelectrolyte hydrogel. Such complexes are stable in water due to the dense interpenetration of the different polymeric chains. The ratio between hydrophilic and hydrophobic functional groups in a gel determines the water-retention capacity and hence the swelling behavior of the system [115]. An advantage brought by a relatively hydrophobic polymer such as poly(lactide-glycolide) (PLG) is that it provides an amphiphilic behavior. On one hand, the overall complex is stabilized by hydrogen bonds between PLG and CS. On the other hand, the hydrophobic characteristics of PLG allow encapsulation of hydrophobic drugs [103]. Apart from this example, the vast majority of polymers used in association with CS are largely hydrophilic and interact with CS via ionic interactions. Table 2 presents a non-exhaustive list of polymers frequently associated with CS, as well as the improved characteristics of interest brought to the system upon non-covalent conjugation. This table also details the presence of other adjuvants and the targeted application field. For example, alginate is a common hydrophilic biopolymer mixed with CS [116,117]. It possesses a similar biocompatibility and low toxicity to CS and undergoes gelation upon addition of calcium ions as adjuvant (a phenomenon sometimes referred to as cross-linking gelation). Small adjuvants frequently used in CS–polymer systems are metal ions, which bring an antibacterial effect and improve the mechanical and thermal properties of the final product [117,118]. As this review focuses on the functionalization of CS, we do not report adjuvants that are application-specific such as delivered drugs, targeted cells, or detected molecules. However, we include application-specific adjuvants that are reported to bring a significant change in the physicochemical characteristics of the mixture. For more details about preparation and application of hydrogels involving CS and other polymers, see reviews from Fu et al., Ahmadi et al., and Pita-Lopèz et al. [119,120,121].

### 3.3. Non-Covalent CS Conjugates with Small Molecules and Nanocomposites

Small negatively charged molecules can be entrapped into CS matrix by electrostatic interactions (also named physical, non-covalent, or ionic cross-linking) with the amino groups of CS chains. Physical properties of the resulting gels such as porosity and solubility can be controlled by changing the concentration, the MW and DD of the CS matrix, the charge density of the ionic agent (e.g., the small molecule adjuvant), or the pH of the medium [119,125,126,127]. Preparation of these gels is performed under mild conditions. Typically, the adjuvant (1–2% *w*/*v*) is added to a CS aqueous solution with 1–2% (*v*/*v*) of acetic acid. Physical cross-linking is considered less prone to toxic side effects than chemical cross-linking due to its sensitivity to medium variations [128]. Furthermore, the small molecular adjuvants presented in this section are non-toxic and already approved as safe additive for human consumption (if applicable). In comparison with covalently grafted molecules presented in the previous section, the presence of unreacted cross-linking agents remaining inside the final material represents potential toxicity.

Citrate is a common small molecule adjuvant used to produce CS gels [125,128,129,130]. Citric acid (CA) is a FDA-approved food-flavoring agent and presents excellent antimicrobial and antioxidant properties [131]. It makes this small molecule a great candidate for bio-oriented applications involving CS. The three carboxyl groups of CA are deprotonated at pH above 6.4 and can form ionic bonds with the amino groups of the CS chains. Citrate addition to a CS solution triggers a gelation process, during which the citrate molecules and the CS chains arrange themselves into an amorphous hydrogel with good swelling and mechanical properties [128]. Alternatively, molecules and NPs with no affinity to CS can be functionalized with citrate in order to increase their stability inside a CS matrix [130,132].

Known for its cross-linking and non-toxic properties, tripolyphosphate (TPP) is very often found in CS hydrogels [126,133] and NPs [134,135]. TPP can have up to five negative charges and is believed to align along the CS chains in a gel, creating sheets of CS-TPP [136]. When working with this complex, the CS:TPP ratio is a key parameter to investigate, as it will influence the stability, rigidity and, if applicable, the size of the NP produced.

CS has also been reported in association with metals, where the complexation of the metal anion with CS is achieved by chelation. The interaction between CS and a metal ion is thought to involve both the amines and the hydroxyl groups of CS chains to form a coordination sphere [137,138]. For example, CS is known to have a strong affinity with Cu^2+^, which was recently explored by Nie et al. [137]. They immerged a mold filled with CS–Cu solution into an alkaline gelation bath and observed the structure of the hydrogel after diffusion of the -OH ions inside the hydrogel. At high Cu^2+^ concentration, CS and metal ions formed strong interactions leading to a dense and rigid multi-layered gel, while lower Cu^2+^ concentration led to a composite gel with oriented fibers (see Figure 8). Calcium anions, on the other hand, are known to have a lower affinity to CS. On the contrary to the CS–Cu^2+^ system, they observed that increasing the concentration of Ca^2+^ did not lead to a change in structure nor a stronger gel. Indeed, the CS–Ca^2+^ gel remained a fibrous material.

In addition to acting as a drug stabilizer and loading matrix, CS also served as a dispersive agent when mixed with nanoparticles and inorganic components. The association of CS with metal NP [139], quantum dots (QD) [140], carbon nanotubes (CNT) [141], and graphene oxides [142] was reported to avoid aggregation of the inorganic adjuvants and reduce their toxicity. In the development of biosensors, porous CS gels were applied to the encapsulation of individual particles [143], resulting in increased surface area for target detection. However, CS-based biosensors are restricted to pH conditions above 6.5 in order to avoid solubilization and detachment from the substrate.

The last category of nanocomposites associated with CS via non-covalent interactions includes natural mineral composites. Minerals coated with biopolymers are used in treatment technologies such as mineral fertilizer, food additives in animal husbandry, pharmaceutics, and cosmetics. Apart from favorable physicochemical and mechanical properties, they provide ion-exchanging and adsorption properties [121]. In treatment technologies, for example, hydroxyapatite–CS material is studied for its capacity to adsorb metal ions from waste water via chelation [144]. This CS–mineral gel was reported to have a higher chelating capacity due to the high density of hydroxyl groups brought to the system by the CS backbone. Kaolin is another example of natural mineral that was incorporated in CS mixture (as particulate additive) for its antibacterial properties [103]. Although the choice of mineral adjuvants is largely application-specific, the use of this type of combinations transforms a homogeneous hydrogel into a heterogeneous mixture.

## 4. Combination of Covalent and Non-Covalent Modifications

In this section, we describe CS-based systems that present both covalent and non-covalent interactions. In these cases, the amino groups of CS units are usually functionalized with a small molecule or a polymer prior to mixing with another component, but a few examples of the reverse process were also disclosed. The resulting systems combine both covalent functionalization patterns, which are stable under a large variety of conditions, and non-covalent interactions, which are more prone to disruption depending on the environment. This dual conjugation strategy offers the opportunity to modulate the stability and hydrophilicity of the final CS-based materials. The main interest in associating the two types of interactions is that it combines the stability of simple covalent grafting with the ease of preparation of self-associating molecules. To the best of our knowledge, hydrophobically-modified CS derivatives were exclusively employed in association with single-walled or multi-walled carbon nanotubes (CNT) [145,146]. Consequently, this section describes almost exclusively hydrophilic systems. The nature of the interactions between functionalized CS and complementary molecules refers mostly to H-bonding and ionic interactions. In some cases, metal ion chelation and aromatic π–π stacking also participate in the stabilization of the system.

### 4.1. Functionalization Strategies of CS Intended to Non-Covalent Mixing

The first step toward dual covalent/non-covalent CS-based materials generally makes use of covalent functionalization of the reactive CS amines. This initial modification of the CS backbone can add valuable properties such as improved solubility and increased bio-adhesion [147] and biocompatibility [75] to the final material. The number of modified glucosamine units is commonly expressed as the grafting degree (GD). As the increased solubility of CS derivatives is known to result from the disturbance of the polymeric chain stacking by grafted molecules, CS amine functionalization is almost always associated with an improved solubility of CS in aqueous solution, independently of the type or role of the molecule grafted (whether sugar, alkyl, or small charged moieties). These conjugation reactions are considered to result in a homogeneous functionalization of the bulk CS polymer (Figure 9A). However, some studies took advantage of an outer-shell derivatization to locally functionalize CS reactive sites. More specifically, when CS was used as a coating agent, derivatization of the polymer was performed after the formation of a core–shell complex (Figure 9B). Shahdeo et al. made use of this strategy to introduce a cancer targeting peptide sequence on CS-coated Au NPs, thus ensuring the accessibility of the bioactive component by selectively anchoring it on the CS amines of the outer shell [148].

CS quaternization was frequently applied to improve the solubility of the polymer in aqueous media by increasing the density of positive charges on the backbone and hence promoting electrostatic repulsions, as well as disturbing the intra- and extra-molecular H-bond network through the incorporation of lateral alkyl chains. Quaternization is commonly achieved through epoxide ring opening, quaternary ammonium substitution, and N-alkylation (See Section 2.1). The recent review by Andreica et al. details synthetic pathways for quaternary ammonium salt CS and physicochemical changes that occur upon its formation [150].

The preparation of amphiphilic CS co-polymers capable of self-assembling into micelles was developed over the last decade. These assemblies offer great promises as vectors for drug and gene delivery. Typically, one or more hydrophilic moieties (such as PEG and sulphate) are first grafted on the CS chain, followed by further functionalization with hydrophobic alkyl or aryl groups. The simultaneous presence of functional moieties and polymer with different water affinities provides the amphiphilic character to the final polymeric material, which assembles into micelles (Figure 10) [151,152]. Applications to pH-responsive nanocarriers were reported in the context of controlled delivery of chemotherapeutics to tumor sites [90,153].

As mentioned in the Introduction, the use and application of CS-based polymers expand. Pre-functionalized CS derivatives such as CMCS or TMC are now also available at a reasonable price from commercial suppliers to ease the work of scientists.

### 4.2. Adjuvants Mixed with Functionalized CS

The properties of CS-based materials can be tuned both by covalent functionalization with small molecules and additional non-covalent combination with secondary components. Below, four categories of adjuvants will be discussed: polymers, small molecular entities, proteins, and nanocomposites.

#### 4.2.1. Chitosan-Based Materials with Polymeric Adjuvants

Table 3 presents studies where CS is stepwise conjugated with a small molecule and further combined with natural or synthetic polymers to bring complementary properties to the final material. Relevant examples include gelatin (enhanced water solubility and water retention) [84], poly(aniline) (electroconductivity) [75], poly(dimethylaminoethyl methacrylate) (DMAEMA), and N,N-methylene bisacrylamide (BisAAm) copolymer (electrostatic interactions) [154]. In particular, gelatin and polyvinyl alcohol are a common combination found for tissue engineering applications. Such dual functionalization is known to increase the rigidity of CS scaffolds, even without covalent modifications [155,156].

The association of two polymers can result in a homogeneous or heterogeneous mixture. In a homogeneous mixture, the non-covalent association of the components involves electrostatic interactions and network inter-penetration. In case the polymeric components do not show similar solubilities in the combination medium or present very different viscosities, one observes the formation of a heterogenous gel. This phenomenon also occurs when air is injected in the solution [158]. Local variation in the properties of a heterogeneous gel was exploited in the work of Shaabani et al. [75] for the successful preparation of a semi-conductive and self-healing scaffold for bone engineering applications. The study involved a ternary mixture composed of a bisguanidine-CS derivative, polyaniline (PANI), and a waterborne polyurethane-based polymer. This mixture allowed the improvement of the solubility of the poorly soluble PANI and the promotion of cell growth and mineralization inside the bone implant scaffold. Electroconductive polymers such as PANI were shown to increase the electric signaling among the cells of interest and improve the microstructure of the scaffolds. The polyurethane-based polymer used in this study demonstrated self-healing properties based on disulfide and gold–thiolate bonds having shape memory effect properties, whose purpose is to provide a solid bone scaffold capable of recovering from cracks due to mechanical strains. The heterogeneous polymeric mixture was further treated with chloroauric acid (HAuCl_4_) and poured into polyethylene molds. The final scaffold showed excellent biocompatibility and mechanical properties and illustrated great potential for the repair of large bone defects.

#### 4.2.2. Chitosan-Based Materials with Small Molecular and Nanocomposite Adjuvants

Small molecules have also been incorporated into functionalized CS gels, allowing further modulation of the material properties by electrostatic interactions (Table 4). While the first covalent conjugation to a small molecule generally improves the solubility of CS by disturbing the packing of the CS chain and/or adding H-bonding sites, the second non-covalent combination with another small molecular entity can bring additional stabilizing chemical bridges. This strategy was applied to CS chains presenting high degree of substitution, which hinders their capacity to form intramolecular interactions. For example, Yang et al. disclosed the use of TPP in combination with a nona-arginine-functionalized quaternized CS to increase the stability of the resulting gel [37]. The secondary molecules generally do not participate in the physical stability of the matrix. They rather act as active cargos (such as painkillers [159,160], antibacterial agents [161,162], or anti-inflammatory drugs [63]) meant to be released over time by diffusion. Fluorescent labels have also been successfully incorporated into functionalized CS for localization [161].

To the best of our knowledge, only two examples relate to non-covalent protection and photoinitiated cross-linkage of CS by small molecules. The first study by Fathi et al. [90] uses sodium dodecyl sulfate (SDS) mixed with CS to temporarily protect the amino groups as ammonium sulfates, followed by controlled maleoylation of the hydroxyls (Figure 11). In a different context, riboflavin was reported as a photoinitiator for cross-linking methacrylate-CS derivatives.

Metal ions, NPs, QD [163], and CNT have also been efficiently incorporated into functionalized CS-based films (Table 5). Their interaction with the functionalized polymeric matrix relies on chelation, entrapment, or π–π stacking. For example, chelating agent such as EDTA [164] and Mel [165] grafted onto a CS backbone are capable of chelating cobalt and copper ions respectively. Encapsulation or coating of NPs is performed when the application involves non-solubilized films or beads. Metal NPs (AuNP [148] and AgNP [84]) and metal oxide NPs (ZnO [63,83] and Fe_3_O_4_ [164,165,166]) were combined with CS-based polymers, providing stabilizing and anti-swelling effects as well as adding complementary features to the resulting conjugates, such as optical contrast, magnetization, or bactericide effect.

More frequently used in bone engineering, mineral compounds such as montmorillonite (MMT) [168,169] and hydroxyapatite [170] have been associated with cross-linked CS-based polymers. CS–MMT conjugates are generally assembled by H-bonding between the CS amines and the MMT surface oxides, and the CS backbone is further cross-linked with small chemical spacers to improve the stability of the resulting gel. The strength of the chelation to CS was reported to be comparable to the ion retention inside the mineral pores [169].

CNT and CS derivatives are conjugated by either π–π stacking interactions [167] or electrostatic interactions when the CNT surface is modified with carboxylic acids [145] (Figure 10). Interpenetration between cross-linked CS derivatives and CNT was also reported [145,146]. These CS–CNT conjugates displayed enhanced dispersity and bioavailability of the material, as well as increased thermic resistance and swelling of the CS-based matrix.

#### 4.2.3. Chitosan-Based Materials with Protein Adjuvants

The immobilization of proteins in CS matrices is generally favored by their large size. However, several parameters such as local charge density and concentration must be finely adjusted to prevent protein misfolding and aggregation, which would impair its biological activity. As depicted in Section 3.1 and Figure 9, proteins can be stabilized by electrostatic interactions with the amino groups of CS derivatives. CS functionalization with small negatively charged spacers (such as maleic acid [161]) was reported to improve the stabilizing and protective effect of CS matrices by providing ionic sites able to interact with both the negative and positive local charges of a protein’s surface. Rusu et al. [161] studied the stabilization effect of BSA by a maleic–CS derivative (MCS). They described that the type of complexation changes with the BSA:MCS ratio in water (Figure 11). Despite maleic functionalization, they found that the zeta-potential of MCS alone remained positive. At high ratio, BSA is in excess, and they observed a clear solution interpreted as a simple polyelectrolyte solution without inter-molecular interactions (Figure 11A). As the ratio decreased, BSA and MCS started to interact, and an opalescent suspension was observed up to the point when the amount of BSA, negatively charged, matched the overall positive charge of MCS (Figure 11B). Finally, excess of MCS led first to a flocculation phenomenon (Figure 11C), followed by a transparent suspension, which indicated the formation of macroscopic insoluble polyelectrolyte complexes (Figure 11D). Overall, coacervates typically reach a maximum turbidity, followed by the formation of a suspension due to the formation of non-stoichiometric BSA/MCS complexes of variable composition.

In addition to electrostatic interactions, entanglement can add to the immobilizing interactions with proteins. For instance, glutaraldehyde–CS conjugates were reported to create intermolecular bridges between the CS chains, thus stabilizing the polymeric network around the protein [166].

While CS is the polymer of interest illustrated in this review, similar functionalization strategies were reported for many other (bio)polymers and nanocomposites, including alg [83], HA [171], CNT [172], cyclodextrin polyester [173], and even proteins [174]. In these examples, non-functionalized CS is mixed to these polymers as adjuvants.

## 5. Characterization of CS-Based Materials

CS-based materials are characterized by a large variety of methods intended to elucidate their molecular composition (chemical characterization), physical structure (physical characterization), and advanced properties (mechanical characterization, behavior in solution) (Figure 12). This section focuses on the characterization of CS derivatives but does not extend to the properties of their conjugates with other components such as molecular payloads and metal ions.

### 5.1. Chemical Characterization

#### 5.1.1. Nuclear Magnetic Resonance

Nuclear magnetic resonance (NMR) is the gold standard technique to assess the molecular composition of covalent CS derivatives as chemical modifications can be detected by specific chemical shifts on the NMR spectra.

For both native CS and its derivatives, ^1^H NMR is usually performed in pure D_2_O [17,51,99] or in a mixture of D_2_O and acidic deuterated solvent to improve CS solubility. Deuterated hydrogen chloride solution (35%) (DCl) [57,73,77] or deuterated acetic acid (AcOD) [34,62] are commonly used. Organic solvents such as DMSO-d^6^ have also been reported [53,57]. Depending on the weight average MW (M_w_) of the polymer and the solvent mixture used, both the shape of the signals and their chemical shifts may vary. In particular, COS are characterized by sharper NMR peaks than higher MW CS samples [64,76]. The water signal is commonly used as reference peak, and its position was reported in the range 4.5–6.0 ppm depending on the solvent mixture used. Moreover, ^1^H NMR is the most common way to determine the DD of CS [17,85] as well as the GD of small molecules [35,77] or polymers [32,39] grafted on CS.

Due to their high MW, CS-based materials are generally only characterized by liquid-state ^1^H NMR. A few examples of ^13^C NMR spectra were nonetheless reported in the literature [64,70,77,95]. The very low solubility of some commercial CS batches in aqueous solutions is an obstacle to their characterizations by NMR analysis before and after derivatization. Nevertheless, with the advances of high field NMR spectrometers, detailed characterization reports were provided for CS–small molecule conjugates and CS-based copolymers [36,37,39,40,41,48,70].

Although quite rare, solid state ^13^C NMR was also used for the analysis of CS derivatives when liquid state NMR was not appropriate [6,55,88,89]. This technique is especially useful for the characterization of CS derivatives designed to be insoluble in common aqueous and organic solvents, such as in the case of films, beads, or membranes for water treatments [6,130].

Along with 1D NMR such as ^1^H and ^13^C NMR, diffusion ordered spectroscopy (DOSY) is a powerful 2D NMR technique that allowsto distinguish covalent and non-covalent CS derivatives. While the 1D NMR spectra of CS conjugates assembled by covalent grafting or non-covalent interactions (e.g., H-bonding) are generally very similar, their 2D DOSY spectra will present different profiles, as only covalently linked systems will show an aligned diffusion pattern. We believe that ^1^H NMR analysis is not sufficient for the characterization of covalent systems, and we suggest the implementation of DOSY NMR in the list of routine analyses for such derivatives. We illustrated the relevance of 2D DOSY NMR in our recent publication describing the functionalization and characterization of CS with small molecule and polymers [41].

#### 5.1.2. Fourier Transform InfraRed Spectroscopy

Due to the fast and simple procedure applicable to both liquid- and solid-state samples, attenuated total reflectance Fourier transform infrared spectroscopy (ATR-FTIR) is probably the most widespread method for chemical characterization of CS derivatives. While the presence of specific functional groups can be probed by IR spectroscopy, the precise molecular composition of CS conjugates is not accessible with this technique and should thus be considered a complement to 1D and 2D NMR analysis. However, FTIR brings relevant information for the characterization of CS-functionalized nanomaterials [103,122,132]. In addition, it can reveal interactions with atoms/ions that are not detected by NMR techniques, such as calcium, zinc, or chromium ions [116,130].

Despite its limitations in characterizing the molecular composition of covalent CS derivatives, FTIR is still given in recent reports as the only analytical technique.

#### 5.1.3. Complementary Chemical Characterizations

Although NMR and FTIR are the principal techniques used for chemical characterization of CS-based systems, they can be associated with other techniques such as Elemental analysis (EA) which has even been used for DD and GD estimation in some cases [65,86,87,88,108]. Moreover UV-Vis spectroscopy can be particularly relevant in the case of aromatic and vinyl containing systems due to their high UV absorbtion [31,65,76]. Such method also provides an additional way to estimate their GD on CS [28,49]. Potentiometric titration has also been reported for GD estimation such as for glycidol grafting [62] or for the preparation of CMCS [28,31].

Mass spectrometry (MS) analyses have been rarely used for CS-based systems, but some examples can be found for matrix-assisted laser desorption/ionization (MALDI) [57], which is the most appropriate MS technique for polymers. However, the analysis of CS-based systems is limited to low to medium MW (≤ 100 kDa), is time-consuming compared to NMR or FTIR, and requires several rounds of optimization.

Lastly, the aldehyde content of oxidized CS was determined by alkalimetry in the work of Keshk et al. providing a way to estimate the extent of the oxidative cleavage of CS units [100].

### 5.2. Physical Characterization

The physical properties of CS derivatives include the determination of their MW, morphology, and stability over various ranges of pH and temperatures.

#### 5.2.1. Gel Permeation Chromatography

Gel permeation chromatography (GPC), which allows us to estimate the MW distribution of a polymer, is well suited to the analysis of linear chains. Therefore, the MW of native CS samples and CS derivatives conjugated to small molecules was efficiently determined by GPC [17,28,74]. In cases of branched systems, such as CS-PEI or CS-PEG copolymers, advanced polymer chromatography (APC) was recently disclosed for a more reliable MW estimation. However, this technique is limited to small MW derivatives [32].

#### 5.2.2. Morphological Characterization

Techniques offering morphological insights into CS composites include X-ray diffraction spectroscopy (XRD), dynamic light scattering (DLS), scanning electron microscopy (SEM), and transmission electron microscopy (TEM).

XRD provides information on the crystalline structure of CS derivatives. Most publications compare the diffraction of native CS and CS composites [32,103,118], as chemical modification of the CS backbone disrupts its crystalline structure: it leads to decreased diffraction peaks [31,43]. As a result, it is often hypothesized that the change of crystallinity might be one of the main reasons for the enhanced solubility of functionalized systems over native CS, due to the destruction of inter- and intramolecular CS hydrogen bonds [32]. Of note, COS is less crystalline than CS, most probably because the smaller chains lead to less hydrogen bonds and thus enhanced solubility in aqueous medium [64]. On the contrary, several non-covalent CS conjugates showed higher crystallinity than native CS [130,175].

In the literature related to CS functionalization, dynamic light scattering (DLS) is used mainly to characterize CS-based NPs intended for drug delivery [86,122] by providing their mean hydrodynamic diameter and zeta potential (surface charge) [37,51,132].

Electron microscopy techniques are also powerful tools for surface and structural characterization of CS derivatives. Scanning electron microscopy (SEM) gives relevant information on the sample’s surface and composition, while transmission electron microscopy (TEM) gives more insights into the inner structure of the sample, such as crystal structure, morphology, and stress state information. An important asset of both techniques is their high resolution: about 0.5 nm for SEM and 50 pm for TEM. SEM can give essential insights into structure porosity, pore size, and roughness degree of hydrogels, nano/micro-fibers, and NPs based on CS [31,43]. The roughness degree is particularly important in applications such as wound healing, where cell attachment and proliferation has to be favored [82,102], or in food packaging, where smooth surfaces are preferred [50]. SEM is also a method of choice to observe and measure the pores of CS composites and molecular organic framework (MOF) [130,175]. Energy-dispersive X-ray spectroscopy (EDX) can be used in addition of SEM to give supplementary chemical qualitative analysis [66,71,80,124]. TEM of CS-based systems is most of the time limited to NPs characterization, and its combination with DLS ensures a complete characterization of their dimensions. TEM can give information on the size of NPs [90,132], the membrane thickness of a CS coating [117], or the presence of a payload inside CS particles [176].

Other methods complementing morphological characterization of CS derivatives are the Brunauer–Emmet–Teller (BET) specific surface area analysis [72,130,175], or atomic force microscopy (AFM) [47,75,79]. They are more rarely encountered in the literature.

#### 5.2.3. Thermal Analysis

Thermal analysis of chemicals follows the evolution and degradation of a sample over increasing temperatures. Two methods can investigate such changes. First, thermogravimetric analysis (TGA) measures over time the mass variation of a sample upon temperature increase. It gives information about physical phenomena, such as phase transitions, absorption, adsorption, and desorption, as well as chemical phenomena including chemisorption, thermal decomposition, and solid–gas reactions (e.g., oxidation or reduction). The evaluation of thermal stability can also be conducted by differential scanning calorimetry (DSC). This method measures the amount of energy required for the sample to undergo specific physical state transitions and does not rely on mass changes as for TGA. As regards CS derivatives, DSC eventually leads to similar information to TGA [100,101,116].

Both covalent and non-covalent CS-based systems show similar TGA diagrams, depending on the temperature range. Below 150 °C, weight loss corresponds to physically adsorbed water or alcoholic solvents. From 250 to 400 °C, one observes the degradation of CS derivatives, resulting from dehydration of the saccharide ring, depolymerization, and decomposition of acetylated and deacetylated units [79]. Above 450 °C, and up to 600 °C, CS derivatives undergo thermal oxidation of the remaining material [88,101]. The difference between the degradation profiles of native CS and CS derivatives accounts for the stabilizing [66,122] or destabilizing [6] effect brought by the modification. However, the rationale behind these variations was not fully elucidated. Similarly to XRD, thermal analysis is relevant for comparison between native CS and CS derivatives.

### 5.3. Mechanical and Rheological Characterizations

Mechanical and rheological properties are largely modified by the addition of other components to CS solutions, gels, or composites [177]. Those modifications are directly application-oriented and aim at improving several parameters such as stability, flexibility, or stiffness of CS-based materials [31,178].

Evaluation of the rheological behavior and viscosity of CS-based systems offers additional insights into their structure. For example, Pan et al. deduced from the storage modulus G’ and loss modulus G’’ of alg/CS hydrogels that the two polymers strongly interacted with each other in an acidic solution thus forming a gel-like structure [116]. This led to hydrogels with low viscosity and high elasticity, which are well adjusted to the needs of food packaging applications. Usually, rheological characterizations for non-covalent systems are performed to evaluate the chain entanglement efficiency [103], whereas for covalent systems, the network density is preferentially investigated [31].

Mechanical parameters commonly evaluated on CS-based materials are Young’s modulus, tensile strength, and elongation at break. They are all extracted from the stress–strain curve obtained after a tensile test and can be finely tuned to match the requirements of specific applications. For example, cellulose nanofiber/CS (CNF/CS) films for dye absorption in water were designed to be tougher than CNF film but more brittle than CS ones in order to confer higher resistance to the water flow and better dyes separating ability [124]. Similarly, Hanafy et al. produced a CS/TiO_2_ NPs/Oleic acid film displaying the correct balance between elasticity (elongation at the break) and tensile strength [101] in order to properly accommodate to the various forces resulting from the patient’s mobility [178].

## 6. Conclusions

CS functionalization can be divided in two main categories: covalent and non-covalent modifications. Covalent functionalization allows for a strong and precise modification of the CS backbone but is sometimes hindered by difficult syntheses and purification processes. On the other hand, non-covalent functionalization provides easy preparation and purification steps, but at the cost of reduced control over the chemical composition and characterization of the final materials. By combining one or both of these approaches, CS-based systems can be prepared to meet many targeted properties. The choice of derivatization of CS greatly depends on the final application as well as the “know-how” of the scientists performing the study. The versatility of CS makes this biopolymer a very promising material for a broad range of applications, and we can expect to see further progress in the preparation and characterization of CS-based materials in the years to come.

## Data Availability

We do not present original data in the manuscript as it is a review article.

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
