# Peer review of "Chitosan Functionalization: Covalent and Non-Covalent Interactions and Their Characterization"

_polymers, 2021, doi:10.3390/polym13234118_

Round 1

Reviewer 1 Report

In this manuscript, the modification of chitosan by means of covalent and non-covalent interactions for the purpose of functionalization is systematically reviewed. The classification of the interactions of chitosan with other molecules is clear, and the aim of modification is also presented. Furthermore, the characterization of chitosan for explaining the modification is also simply discussed. This review accords with the scope of Polymers. This manuscript is well organized, and can be accepted for publishing after minor revision.

Please note the following minor issues:

The list of abbreviations in the end is suggested.

Please note the use of English tenses in some sentences.

Abstract:

Lines 10: “ Abstract: Chitosan (CS) is a natural biopolymer that gained great interest ---”: “that gained” should be replaced by “that has gained”.

Strategies for covalent functionalization:

Lines 76-77: “The common methodologies --- relies on ---”, Grammar error.

Non-covalent modifications:

Lines 413-415: “Strong of this observation, Wu et al. studied oral and intramuscular delivery of silica NP loaded with BSA (as antigen model) coated with CS [110]”: This sentence should be improved.

Lines 433-434: “On one hand, hydrophilic affinity of PLG with CS chains stabilize the overall polyplexes”: This sentence should be improved. Additionally, “hydrophilic affinity of PLG with CS chains” is not a clear description.

Line 436: “the many polymers used in association with CS are ---”: This sentence should be improved.

Lines 440-441: “For example, alg is a common hydrophilic biopolymer mixed with CS [116,117]”: The full name of alg should be given.

Combination of covalent and non-covalent modifications:

Line 591: “As mentioned in the Introduction, the use and application of CS-based polymers expands”: “expands” should be replaced by “expand”.

Lines 604-605: “Poly(dimethylaminoethyl methacrylate) (DMAEMA)”: “Poly” should be replaced by “poly”.

Line 606: “In particular, gelatin and polyvinyl alcohol is a common ---”: “is” should be replaced by “are”.

Lines 678-679: “Montmorillonite” should be replaced by “montmorillonite”.

References:

Refs. 5, 6, 8, 12, 13, 21, 22, 24, 27, 38, 41, 43, 45, 48, 52, 53, 54, 57, 59, 66, 67, 68, 69, 79, 81, 86, 87, 88, 92, 102, 118, 123, 131, 148, 160, 161, 168, 170, 174, and 176: Please note the case sensitivity (small and capital letters) of paper titles.

Refs. 22. Polymers, 68. Polymers (Basel), 87. Polymers (Basel), 118. Polymers (Basel), 141. Polymers (Basel), 160. Polymers (Basel), 170. Polymers (Basel): Delete “(Basel)”.

Author Response

Please see enclosed document.

Reviewer 2 Report

Reviewer Comment for Editor/Editor-in-Chief and/or Authors:

This review manuscript provides a study focusing on the last five years literature of Chitosan (CS) functionalization (i.e. covalent and non-covalent) as well as the characterization techniques for CS derivatives.

It is a good review manuscript. Well written and organized. It is basically OK and could potentially be suitable for publication after very minor revisions

  1. Page 3, line 80, CMCS, TMC and GCS should be fully identified.

  1. Page 12, line 404, Also DLS and ζ-symbol should be identified.

  1. Page 14, line 473, the authors claimed that” Citric acid (CA) is a FDA-approved food-flavoring agent and presents excellent antimicrobial and antioxidant properties”. Please add citation to support this.

  1. Page 15, line 512, I think the term “non-functionalized” should be “non-covalent functionalized”. Isn’t it?

  1. References should be revised carefully as well as uniformly formatted. For example, please have a look to ref. 1, 2 and 5, these references have need to be revised compared to other references i.e. page number. Also, Ref. 14 is written wrongly.  

Author Response

Please, see enclosed document.
